# Sorting and Manipulation of Human PGC-LC Using PDPN and Hanging Drop Cultures

**DOI:** 10.3390/cells11233832

**Published:** 2022-11-29

**Authors:** Brahim Arkoun, Pauline Moison, Marie-Justine Guerquin, Sébastien Messiaen, Delphine Moison, Sophie Tourpin, Christelle Monville, Gabriel Livera

**Affiliations:** 1Laboratoire de Développement des Gonades, UMRE008 Stabilité Génétique Cellules Souches et Radiations, Université Paris Cité, Université Paris-Saclay, CEA, 92265 Fontenay-aux-Roses, France; 2INSERM U861, I-Stem, AFM, Institute for Stem Cell Therapy and Exploration of Monogenic Diseases, 91100 Corbeil-Essonnes, France; 3Paris-Saclay Evry, U861, 91100 Corbeil-Essonnes, France

**Keywords:** Induced pluripotent stem cells, primordial germ cells, human germline, meiotic commitment

## Abstract

The generation of oocytes from induced pluripotent stem cells (iPSCs) was proven efficient with mouse cells. However, no human iPSCs have yet been reported to generate cells able to complete oogenesis. Additionally, efficient sorting of human Primordial Germ Cell-*like* Cells (hPGC-LCs) without genomic integration of fluorescent reporter for their downstream manipulation is still lacking. Here, we aimed to develop a model that allows human germ cell differentiation in vitro in order to study the developing human germline. The hPGC-LCs specified from two iPS cell lines were sorted and manipulated using the PDPN surface marker without genetic modification. hPGC-LCs obtained remain arrested at early stages of maturation and no further differentiation nor meiotic onset occurred when these were cultured with human or mouse fetal ovarian somatic cells. However, when cultured independently of somatic ovarian cells, using BMP4 and the hanging drop-transferred EBs system, early hPGC-LCs further differentiate efficiently and express late PGC (DDX4) and meiotic gene markers, although no SYCP3 protein was detected. Altogether, we characterized a tool to sort hPGC-LCs and an efficient in vitro differentiation system to obtain pre-meiotic germ cell-*like* cells without using a gonadal niche.

## 1. Introduction

Many efforts have been made to reproduce human oogenesis in vitro; however, this goal remains elusive, while oocyte-like cells were successfully obtained from murine pluripotent cells. Oogenesis is a complex cellular and molecular process by which fertilizable oocytes are produced from primordial germ cells (PGC). Human oogenesis is initiated early during development, although it becomes functional only after puberty. All events which lead to the formation of the pool of follicles that form the ovarian reserve occur during fetal life. This noticeably limited our understanding of the mechanisms governing the establishment of the human female germline. Thus, much of our knowledge about PGC specification and differentiation originate from mouse models. However, human PGC appear to differ from murine PGC due to their highly asynchronous development and the fact that they acquire the ability to express meiotic genes, an event termed gametogenic competency, much later. Indeed, we and others observed that human oogonia coexist for several months with cells initiating and completing meiosis prophase I in the human ovary [1,2,3,4]. Deleted in Azoospermia Like (DAZL) and Dead-box Helicase 4 (DDX4) are two RNA-binding proteins specific to the germline, and DAZL was proven mandatory to allow the expression of the meiotic program in mouse PGCs [5]. Both appear in mouse germ cells just after gonadal colonization, while in human oogonia these factors are reported to appear only weeks after PGC have reached the fetal ovary [6]. Several lines of evidence attributed meiotic initiation to retinoic acid (RA) that is able to induce the expression of a key gene to meiotic onset, *Stimulated by Retinoic Acid 8 (STRA8)* [7] despite its debatable physiological involvement [8]. In addition to RA signalling, Bone Morphogenetic Protein (BMP) contributes synergistically to specify female germ-cell fate in mice and meiotic prophase entry [9].

Considerable efforts have been made to enable pluripotent stem cells to differentiate into oocytes. Such work was proven efficient with mouse induced pluripotent stem cells (iPSCs). Imamura et al. [10] demonstrated that meiotic oocytes can be differentiated from murine iPSCs. This required the formation of embryoid bodies (EB) and exposure to RA, BMP4 and Stem Cell Factor (SCF). Hayashi et al. [11] demonstrated that PGC-like cells (PGC-LC) induced from murine iPSCs are able to complete oogenesis once transferred in vivo and can give rise to healthy and fertile offspring. A few years later, the same team showed near-complete in vitro oogenesis reconstitution from murine iPSCs [12]. The generated oocyte-*like* cells gave rise after fertilization to healthy and fertile offspring. These models rely on the use of fetal female gonadal somatic cells, the ovarian niche, to instruct early PGC-LC differentiation. Human iPSCs have not been reported to generate cells that have the ability to complete oogenesis. It was shown that hiPSCs with a naive pluripotency state, obtained in vitro in the presence of 4 inhibitory small molecules of kinases [13], are more competent to give rise to hPGC-LCs compared with conventional iPSCs [14]. This study allowed the identification of the *SOX17* transcription factor as a specific regulator of human PGC [14]. Recently, it was reported that hPGC-LCs specified from iPSCs failed to complete meiosis in the absence of the supporting gonadal niche [15]. Another team showed that hPGC-LCs produced from human iPSCs can progressively differentiate into oogonia-*like* cells during long-term in in vitro culture (approximately four months) in xenogeneic reconstituted ovaries with mouse fetal ovarian somatic cells [16].

Profound differences in the development of the mouse and human germline may justify difficulties encountered to establish a robust model that would allow human germ cell differentiation in vitro. Additionally, isolation and manipulation of PGC typically requires the use of transgenic reporters based on endogenous markers such as TFAP2C or BLIMP1 [16]. Such modification in the human genome may not be wanted. Few cell surface markers of PGC were proven efficient to sort these cells, apart from SSEA1 in the mouse. The transmembrane protein podoplanin (PDPN) is known to be expressed in several embryonic tissues including germ cells [17]. It has been shown that its expression in germ cells was exclusive from DDX4, therefore it can be used to distinguish early from late PGC in the developing gonad [18]. Here, to prevent genetic manipulations, we propose the PDPN as an alternative marker of early PGC, targetable for downstream processing. Also, using various culture methods, including aggregate formation, we assess the potential of hPGC-LCs to develop further. Notably, we set up an in vitro differentiation system to obtain DDX4-positive cells without using reconstituted aggregate with endogenous gonadal cells.

## 2. Materials and Methods

### 2.1. In Vitro Culture of Naïve-like hiPSCs

Human iPS cell lines (PB12.CO3: 46XX; derived from Peripheral Blood Mononuclear Cells and SMA: 46XX; derived from fibroblasts of a safe mother whose child was affected by Spinal Muscular Atrophy) were grown on 10 µg/mL Mitomycin C- (Sigma Aldrich, St Quentin Fallavier, France) treated Mouse Embryonic Fibroblasts (MEFs) in primed pluripotency conditions. These culture conditions contained Ham’s F12/Dulbecco’s Modified Eagle’s Medium (DMEM/F12), 1× GlutaMax, 20% Knock-Out Serum Replacement (KSR) (All GIBCO, Villebon sur Yvette, France) and 10 ng/mL b-FGF (PeproTech, Neuilly-Sur-Seine, France). Media were refreshed every day and the cells were mechanically passaged every seven days in the presence of 10 µM ROCK inhibitor (Y-27632, Sigma, St Quentin Fallavier, France) for 24 h. To derive naive-*like* iPSCs, primed iPS cell lines were cultured and maintained in Naive Human Stem Cell Medium (NHSM) as previously described [13]. Briefly, cells were grown on Mitomycin C-treated MEFs in knockout DMEM supplemented with 20% KSR, 2 mM L-glutamine, 0.1 mM nonessential amino acids, 0.1 mM 2-mercaptoethanol (all GIBCO, Villebon sur Yvette, France), 20 ng/mL human LIF, 8 ng/mL bFGF, 1 ng/mL TGF-b1 (All Peprotech, Neuilly-Sur-Seine, France), 3 µM CHIR99021, 1 µM PD0325901, 5 µM SB203580, and 5 µM SP600125 (All Biotechne, Rennes, France). Cells were passaged every 3 to 5 days using Accutase (GIBCO). 10 µM of ROCK inhibitor (Y-27632, Sigma) was used for 24 h after the passage.

### 2.2. In Vitro Differentiation of Early and Late hPGC-LCs

hPGC-LCs were specified from naive-*like* hiPS cell lines as previously described [14]. Briefly, cells were plated in ultra-low attachment U-bottom 96 well plates (Corning, New York, USA) at a density of 8000 to 10,000 cells/well in a PGC-LC medium. The latter was composed of Glasgow’s Minimal Essential Medium (GMEM, GIBCO), 15% KSR (GIBCO), 0.1 mM nonessential amino acids (GIBCO), 0.1 mM 2-mercaptoethanol (GIBCO), 100 U/mL Penicillin- 100 U/mL Streptomycin (GIBCO), 2 mM L-Glutamine, 1 mM Sodium pyruvate (Sigma), and the following cytokines: 500 ng/mL BMP4 (Peprotech), 1 µg/mL human LIF (Peprotech), 100 ng/mL SCF (Peprotech), 50 ng/mL EGF (Peprotech), and 10 µM ROCK inhibitor. To force EB aggregation, cells were centrifuged at 200× *g* for 2 min and cultured for 4 days. To further differentiate early hPGC-LCs, day 4 EBs were transferred in hanging drops (one EB per hanging drop) for an additional 6 days in a medium composed by DMEM/F12 (GIBCO), 15% KSR (GIBCO) containing the following cytokines: 100 ng/mL BMP4 (Peprotech) and 20 ng/mL SCF (Peprotech). Day 4 EBs and those transferred to hanging drops were exposed or not to 1 µM retinoic acid (Sigma Aldrich, St Louis, USA).

### 2.3. Collection of Human Fetal Ovary

Human fetal ovaries were harvested from material available following legally induced abortions in the first trimester of pregnancy, i.e., aged from 7.8 to 10 gestational week (GW), in the Department of Obstetrics and Gynecology at the Antoine Béclère Hospital, Clamart (France) as described before [4,19]. This study was approved by the Biomedicine Agency (authorization number PFS12-002) and all women gave their informed consent. Fetus ovaries were recognized and selected according to the morphology of the gonads and their age was determined by measuring the length of limbs and feet [20].

### 2.4. Collection of Mouse Fetal Ovary

The animal studies were conducted in accordance with the French Ministry of Agriculture guidelines for the care and use of laboratory animals. Fetal ovaries from POU5F1-GFP mice [21] were harvested at 12.5 days post-conception (dpc). Briefly, males and females were caged together overnight, and the presence of a vaginal plug was examined the following morning. Pregnant females were sacrificed by cervical dislocation and their fetuses were removed from the uterine horns. The gonads were devoid of mesonephros and enzymatically digested as previously described [22].

### 2.5. Real-Time Quantitative RT-PCR

Total RNA was extracted using the RNeasy minikit (QIAGEN, Valencia, CA, USA) and reverse transcription was carried out using the high capacity kit (Applied Biosystems, Foster City, CA, USA) according to the manufacturer’s instructions. The 7900HT Fast Real-Time PCR System (Applied Biosystems) and SYBR-green labeling were used for quantitative RT-PCR (RT-qPCR). The comparative ΔΔcycle threshold method was used to determine the relative quantities of mRNA using ACTB (ß-actin) mRNA as reference gene for normalization. Each RNA sample was analyzed in triplicate. The results are presented in histograms showing the relative expression of the studied marker with standard error of mean (sem). The sequences of oligonucleotides used with SYBR-green detection were designed with Primer Express Software and are illustrated in Appendix A.

### 2.6. Single Cell RNA Seq

Two sets of 25 EBs at day 4 were washed in PBS and dissociated with TrypLE Express for 30 to 60 min at 37°c with repeated flushing, and the reaction was stopped with medium plus KSR, and cells were centrifuged at 300× *g* for 15 min and resuspended into PBS-BSA 0.05%. Single-cell suspensions from day 4 EBs were pooled before Hoescht 33258 staining. Total viable cells were flow-sorted, the cells were counted and 4.000 were loaded onto the Chromium 3′ chip. Reverse transcription, library preparation and sequencing were performed according to the manufacturer’s recommendations (10× Genomics, Pleasanton, CA, USA). Two replicates were performed for the scRNAseq study. Samples were sequenced on a single run with the NovaSeq 6000 system (Illumina, San Diego, CA, USA). Data were preprocessed with Cell Ranger software v6.1.2 from 10× Genomics. For analysis of scRNA-seq data, Seurat package v4.0.4 was used in R version 4.1.1 [23]. Separate preprocessed data sets were merged into one Seurat object and analyzed. Data were log-normalized and scaled. Principal component analysis (PCA) was performed based on the 3000 most variable genes, selected using the vst method and 20 PCA dimensions were used for graph-based clustering with a resolution of 0.8. Clusters were visualized with a UMAP algorithm. When zooming in, 20 PCA dimensions were used for graph-based clustering. A cluster of cells with higher level of mitochondrial genes, low count of genes and no POU5F1 expression (cluster 3) was removed before reclustering using a resolution of 0.5 and 20 PCA dimensions. The cluster of PGC-like cells was used for subclustering with a resolution of 0.1 and using 20 PCA dimensions. A co-expression correlation of BMP receptor type I and II was performed using the sum of log-normalized expression of *BMPR1A* plus *ACVR1* for type I BMP receptors and *BMPR2* plus ACVR2B for type II BMP receptors. The co-expression correlation of Retinoic Acid receptors and *CYP26A1* was performed using the sum of log-normalized expression of *RARA*, *RARB* plus *RARG*.

### 2.7. Immunohistochemistry

Immunohistochemical staining using appropriate antibodies (Appendix A) was performed as follows. Tissue sections were mounted on glass slides, dewaxed, rehydrated and submitted to antigen retrieval by boiling for 20 min in a citrate buffer (pH 6). Endogenous peroxidase activity was blocked by a 10 min incubation with 3% hydrogen peroxide. Slides were then washed in PBS and blocked for 30 min in 2% horse serum. Slides were incubated overnight at 4 °C with primary antibody diluted in PBS, 20% blocking buffer. Bound primary antibody was revealed by 30 min incubation with peroxidase-conjugated secondary antibody (ImmPRESS reagent kit, Vector Laboratories) and finally with 3,3′-diaminobenzidine (DAB substrate reagent kit, Vector Laboratories). For immunofluorescence, 4% formaldehyde-fixed embryoid bodies (EB) or cell mixture aggregates (CMA) were submitted to antigen retrieval with citrate buffer (pH 6). Sections were then blocked either in 2% horse serum, and the slides were mounted with the Vectashield 4,6-diamidino-2-phenylindole (DAPI) medium (Vector Laboratories, Newark, CA, USA). Imaging was performed using a Leica DM5500 B epifluorescence microscope (Leica Microsystems, Wetzlar, Germany) equipped with a CoolSNAP HQ^2^ camera (Photometrics) and Leica MMAF software (Metamorph). Images were processed with Image J software.

### 2.8. Fluorescence-Activated Cell Sorting

Day 4 EBs were washed in PBS and dissociated with TrypLE Express for 30 min at 37 °C. Human or mouse fetal ovaries were dissected out from surrounding somatic tissues in PBS and dissociated with TrypLE at 37 °C for 15–30 min. Dissociated cells were resuspended in a solution consisting of 3% (*v*/*v*) fetal bovine serum (FBS, GIBCO) in PBS. EBs and human ovaries were incubated with anti-PDPN antibody at 1/20 (DAKO, Santa Clara, CA, USA) for 30 min, then with anti-mouse antibody conjugated with Alexa Fluor 488 (Thermofisher, Waltham, MA, USA). No antibody staining was performed for cells from mouse fetal ovaries since POU5F1-positive germ stem cells already express the Green Fluorescent Protein (GFP). After washing with PBS, the cells were treated with 1 mg/mL DAPI and were sorted by BD FACS ARIA III (BD Biosciences, Franklin Lakes, NJ, USA).

### 2.9. Aggregate Cultures

After cell sorting, two types of CMAs were reconstituted from: (i) day 4 EBs primordial germ cell-*like* PDPN-positive cells with human fetal ovarian PDPN-negative somatic cells, or (ii) day 4 EBs primordial germ cell-*like* PDPN-positive cells with mouse ovarian POU5F1-GFP negative somatic cells at a ratio of 1:8 (1 PGCLCs for 8 human/mouse ovarian somatic cells). In order to force aggregation, the cell mixtures were plated in ultra-low attachment 96 well plates (Corning) and centrifuged at 200× *g* for 2 min. After 48 h, CMAs were transferred to hanging drops in a culture medium adapted from Hikabe et al. [12] and composed of alpha-Minimal Essential Medium (αMEM, GIBCO), 10% SVF (GIBCO), 1.5 mM ascorbic acid, 10 U/mL Penicillin, and 10 U/mL Streptomycin (GIBCO).

### 2.10. Statistical Analyses

A statistical analysis was performed for all experiments using the Mann–Whitney test for unpaired rank comparisons. The data were presented as the mean ± s.e.m.; and a *p*-value below 0.05 was considered to be statistically significant.

### 2.11. Data Availability

Sequencing data and processed expression matrices have been deposited at the European Bioinformatics Institute (EMBL-EBI) under the accession numbers E-MTAB-12089 (single-cell RNA-seq).

## 3. Results

### 3.1. The In Vitro Specification of hPGC-LCs from Naïve-like hiPSCs

To optimize hPGC-LC specification, naive-like hiPSCs were derived from conventional/primed hiPSCs using the Naive Human Stem Cell Medium, as described before by Gafni et al. [13] (Figure 1A). After 15 days of derivation, the large-flattened conventional iPSCs changed into compact-domed colonies corresponding to a characteristic morphology for naïve-like hiPSCs [24] (Figure 1(B1)). hPGC-LCs were induced in vitro from naïve-like hiPSCs via embryoid bodies formation during 4 days of culture in the presence of BMP4, LIF, SCF and EGF (Figure 1A,B2). In order to situate these cells in their differentiation state, gene expression in the embryoid bodies was compared to human fetal ovaries aged from 7.8 to 10 GW. Day 4 EBs showed the maintenance of pluripotency genes (*POU5F1* and *NANOG*, Figure 1C) with a significant increase of early PGC genes (*SOX17*, *CD38*, *TFAP2C*, *NANOS3*, *KIT* and *PDPN*) when compared with 4i hiPSCs (Figure 1D). However, late PGC genes, such as *DAZL* and *DDX4* (also known as *VASA*), were only detected in the human fetal ovary (Figure 1E). Similarly, no expression was detected for the meiotic genes *STRA8* and *SYCP3* in day 4 EBs. *MEIOC* and *DMC1*, two other meiotic genes, displayed only a slight increase when compared with 4i hiPSCs (Figure 1F). Since it was proposed that RA regulates meiotic onset in both spermatogenesis and oogenesis [25], we wanted to assess whether RA treatment affects hPGC-LC differentiation. Surprisingly, RA did not influence the expression levels of pluripotency, early and late PGC nor that of meiotic genes when compared with day 4 EBs without RA (Figure 1C–F). Altogether, these data show that early hPGC-LCs can be specified from naïve-*like* hiPSCs via day 4 EB formation in vitro despite the fact that those cells lack gametogenic competency.

### 3.2. Single-Cell Transcriptomic Analysis of PGC-LC Reveals Active BMP-Signaling

To obtain a precise understanding of the molecular features in the generated early PGC-LC, scRNA-seq of day 4 EB cells was performed from two series of experiments. A total of 5411 cells sharing similar characteristics in the number of genes, UMIs and mitochondrial genes were analyzed (Appendix A). UMAP clustering obtained by pooling cells of each experiment provided nine clusters (Appendix A). To distinguish PGC-LC from germ layers, scores for lineage-specific gene lists were computed (Appendix A and Figure 2A,B).

Focusing our analysis on the germ cell fraction, three sub-clusters of PGC-LC were identified and overexpressed early PGC markers such as *TFAP2C*, *POU5F1* and *NANOG* compared to somatic cell-like cells (Figure 2C,D). As *bona fide* PGCs undergo dramatic DNA demethylation that establishes the germline epigenetic ground state, we sought to verify the status of a critical regulator of DNA demethylation in female germ cells, TET1. TET1, that converts 5-mC to 5-hmc for DNA demethylation, was overexpressed in the three sub-clusters of PGC-LC compared to somatic cell-like cells suggesting that these PGC-LC are in a DNA hypomethylated state. TET1 expression progressively increased in PGC-LC from cluster 1 to 3 (Figure 2C,D). Germ cell markers known to be induced upon gametogenic competency and meiotic entry were not expressed such as: *DAZL*, *DDX4* and *SYCP3* (Appendix A), consistent with the early PGC state of these cells in day 4 EBs.

Although rare hPGC-LC surface markers were identified such as *CD38* [14], difficulties to sort germ cells are still encountered and alternative surface markers are lacking. It was previously reported that PDPN distinguishes early PGC from gonadal somatic cells in humans and marmosets [26,27,28]. ScRNA-seq indicates that *PDPN* was strongly expressed and *CD38* was very weakly detected in PGC-LC (Figure 2E). *PDPN* mRNA was retrieved in the majority of PGC-LC co-expressing *SOX17*, a PGC marker (Figure 2F) and in some somatic cell-like cells (Appendix A). Our results are thus consistent with *PDPN* being expressed in PGC and PGC-LC.

Since BMP signalling is key to reaching the later stages of the PGC differentiation program prior to meiotic entry both in vivo and in vitro with mouse embryonic progenitors [9], we sought to determine whether our early PGC-LC exhibits molecular features of BMP signalling. BMPR1A and ACVR2B genes encoding BMP receptors forming a heteromeric complex are overexpressed in PGC-LC compared to somatic cell-like cells (Figure 2G). Approximately 20% of PGC-LC co-expressed high levels of Type I and II BMP receptors compared to only 1% of somatic cell-like cells (Appendix A). In addition, Type I BMPR is positively correlated to *SMAD5* expression, its transcriptional effector, in PGC-LC (Appendix A), indicating a likelihood of active BMP signalling in these cells. In fact, BMP signalling was especially prominent in some PGC from cluster 3 which also harbour cells with high levels of *TET1* and low pluripotency markers. Surprisingly, the mRNA of RA receptors corresponding to *RARA*, *RARB* and *RARG* genes that were described to be crucial for meiotic entry were dramatically downregulated in PGC-LC compared to somatic cell-like cells (Figure 2H). Subsequently, hardly any PGC-LC exhibited co-expression of RARs with CYP26A1, a well-known RA-target gene (Appendix A) indicating that RA signalling is likely inefficient in hPGC-LC.

Altogether, our results strongly suggest that day 4 PGC-LC harbour the transcriptional features of *bona fide* early PGC and are able to respond to BMPs, consistent with further differentiation potential.

### 3.3. Detection of Early PGC Protein Markers, TFAP2C and PDPN, in Day 4 Embryoid Bodies

To verify the presence of early PGC markers at the protein level, an immunohistochemistry (IHC) was performed. The TFAP2C protein was detected in day 4 EBs in the absence or presence of RA (Figure 3A). Stereological analyses showed more than 30% of TFAP2C-positive cells per day 4 EB section and similar results were obtained with RA-treated EBs (Figure 3B). To verify the proliferation potential of these early hPGC-LCs, the expression of the Ki67 marker was assessed (Appendix A). More than half of the TFAP2C-positive cells were also Ki67-positive indicating that hPGCLC divide. Additionally, 72.5% of TFAP2C-positive cells co-expressed the pluripotency factor POU5F1 (Figure 3C), reinforcing their likely PGC identity. In order to isolate PGC-LC, we searched for a specific membrane-bound antigen and therefore assessed PDPN, which was detected on the mRNA level through scRNA-seq analysis. In day 4 EBs, the PDPN protein was detected at the surface of cells resembling hPGC-LCs by IHC (Figure 3D). Of interest, more than 90% of PDPN-positive cells were also stained for POU5F1, confirming the PGC-like identity of these cells (Figure 3E). On the other hand, very few somatic-like cells presented a staining for PDPN (1.01 ± 0.20% of POUF1-negative cells were PDPN-positive, *n* = 4), indicating a specific presence of the protein at the membrane of hPGC-LC and contrasting with scRNA-seq data (62.8% of somatic-like cells expressing the *PDPN* mRNA, Figure 2E). The same PDPN antibody was used for hPGC-LCs sorting from day 4 dissociated EBs. Similar to stereological analyses of TFAP2C, 31% of PDPN-positive cells were obtained with fluorescent activated cell sorting (Figure 3F). These results provide a new tool to sort and manipulate early hPGC-LCs using the PDPN surface marker.

### 3.4. hPGC-LCs Remain Arrested at Early Stages When Cultured with Human Ovarian Somatic Cells

To assess if the human ovarian somatic environment can enhance the differentiation of day 4 hPGC-LCs, fetal somatic ovarian cells of 8 GW were mixed with PDPN-positive hPGC-LCs. Since it was already known that seven to eight GW ovaries contain a ratio of one oogonium for eight somatic cells [29], the two populations of PDPN-positive hPGC-LCs and PDPN-negative somatic ovarian cells were sorted and aggregated at a ratio of 1:8 (1 PGC-LCs for 8 human ovarian somatic cells; hu-L/hu) (Figure 4A,B). Aggregation was performed in a U-bottom 96-well plate (at 100× *g*, 2 min) for 48 h before the transfer of the newly formed Cell Mixture Aggregates (CMA) in hanging drops up to 15 or 30 days. A MOCK (only somatic cells), devoid of any hPGC, was also cultured to ensure the absence of any contamination from ovarian germ cells (Figure 4B). In order to validate the efficiency of our culture system to allow early hPGC maturation, a fraction of the somatic ovarian cells was re-assembled with PDPN-positive cells (hPGCs, hu/hu) from the same ovary (Figure 4B). At day 15, immunofluorescence analysis showed the complete absence of TFAP2C and DDX4-positive cells in the MOCK condition (Figure 4C). Late DDX4-positive hPGC were detected in the CMA (hu/hu) with no TFAP2C protein presence in those cells (Figure 4C). In CMA-containing PGC-LCs (hu-L/hu), TFAP2C-positive cells were still present while no DDX4-positive cells were detected (Figure 4C). However, culture up to day 30 of CMA (hu-L/hu) did not lead to the appearance of the DDX4-protein in hPGC-LCs that still remained in TFAP2C-positive cells (Figure 4D). Altogether, these data show a long-term persistence of hPGC-LCs in CMA, although these remain at an early stage of maturation without further differentiation when cultured in a human ovarian somatic environment.

### 3.5. The Somatic Environment of Mouse Fetal Ovary Is Unpermissive for hPGC-LC Maturation and Meiotic Onset

Unlike in humans, in mouse fetal ovaries, all germ cells undergo meiosis in a synchronous manner with no maintenance of undifferentiated PGCs [4,7,30]; this may thus represent a niche more permissive to meiotic entry. To assess if the mouse somatic environment allows further differentiation of hPGC-LCs, day 4 EBs and mouse fetal ovaries (MFO) were processed as described in Figure 5A. Pou5f1-GFP mice that specifically express the GFP in PGCs were used to sort germ and somatic cells from the ovaries. Three types of aggregates were formed: (i) MOCK constituted only by mouse GFP-negative somatic cells (ii) CMA (ms/ms) corresponding to a reconstituted mouse fetal ovary, with both GFP-positive and -negative cells, used as a positive control and (iii) CMA (hu-L/ms) constituted by PDPN-positive hPGC-LCs and GFP-negative (somatic) mouse ovarian cells at a 1:8 ratio (Figure 5B). Two days after the low adherence culture, the cell mixtures formed aggregates and were transferred for 6 or 14 additional days in a hanging drop culture system (Figure 5C). Immunofluorescence analyses revealed the presence of DDX4 and SYCP3 proteins on day 8 CMA (ms/ms), while they were completely absent in MOCK negative control (Figure 5D). At day 16 of culture, CMA (ms/ms) showed the formation of primary follicles containing oocytes at the diplotene stage associated with the expression of DDX4 but no SYCP3 protein, while no germ cells were found in the MOCK condition (Appendix A). This indicates an efficient meiotic entry and progression in the mouse ovarian niche. However, in CMA (hu-L/ms) at day 8, no further differentiation of early hPGC-LCs was observed as neither DDX4- nor SYCP3-positive cells were detected. These cells retained the TFAP2C marker until day 16 (Figure 5E). To ensure that the early hPGC-LCs blockade is not related to the genetic background of the SMA hiPSC line, the same experiment was repeated using the PB12 hiPSC line. Similarly, DDX4 and SYCP3 proteins were not detected in PB12 CMA (hu-L/ms) at day 8 of culture (Appendix A). The long-term culture, until day 16, of PB12 CMA (hu-L/ms) allowed neither the expression of DDX4 nor SYCP3 proteins (Appendix A). In order to validate the efficiency of the culture system used with human PGCs, PDPN-positive cells from human fetal ovaries were mixed with mouse ovarian somatic cells as described previously. In these CMA (hu/ms), hPGCs differentiate and showed SYCP3 meiotic marker expression at day 8 of culture (white arrowheads of Figure 5F). In order to confirm the human origin of these cells, two different SYCP3 antibodies were used; one recognizing only the mouse SYCP3 and the second one recognizing both mouse and human SYCP3 proteins as shown in positive control ovaries (Figure 5F). Altogether, these data show that early-stage hPGC-LC are intrinsically poorly competent to further maturate but can be maintained with long-term persistence in a murine or human ovarian somatic environment.

### 3.6. In Vitro Overexpression of Late PGC and Meiotic Gene Markers in Hanging Drop Transferred-EBs

In an attempt to further differentiate early hPGC-LC, we set up an alternative strategy by transfering day 4 EBs into hanging drops offering the advantage to the cells to be in a well oxygenated three-dimensional micro-environmental niche. Since our scRNA-seq data (Figure 2G) showed that early hPGC-LC are able to respond to BMPs and that BMP4 was already shown to promote germ cell differentiation, we decided to supplement our culture medium with BMP4 (Figure 6A,B). After 6 and 8 days of culture in the hanging drop, EBs displayed a decreased expression of pluripotency genes such as *POU5F1* and *NANOG* when compared either with 4i hiPSCs (Appendix A) or day 4 EBs (Appendix A). Although hanging drop EBs maintained high expression levels of early PGC genes such as *SOX17*, *CD38* and *NANOS3* when compared with 4i hiPSCs (Appendix A), these were decreased when compared with day 4 EBs (Appendix A). On the contrary, the expression of PDPN increased when compared with day 4 EBs (Appendix A). Interestingly in hanging drop cultures after 6 days, the expression of late PGC genes such as DAZL and DDX4 increased (*p* = 0.02; for both) when compared with day 4 EBs or 4i hiPSCs (Figure 6C and Appendix A). Similarly, the expression of the meiotic genes investigated increased robustly (Figure 6D and Appendix A) at days 6 and 8 in the hanging drop. Additionally, the expression of synaptonemal complex genes significantly increased at day 6 in the hanging drop when compared with 4i hiPSCs or day 4 EBs (Appendix A). RA treatment for 48 h (days 6 to 8) did not modify the expression of *DDX4*, *STRA8*, *MEIOC*, and *DMC1* genes. However, RA prevented the decreased expression of *DAZL* and *SYCP3* that were mildly decreased in +D8 HD when compared with +D6 HD (Figure 6C,D and Appendix A). Altogether, these data show that the hanging drop culture system allows the further differentiation of hPGC-LCs with transcriptional levels of late PGC and meiotic genes expression similar to the human fetal ovary control (7.8 to 10 GW). Since little difference was observed between +D6 HD and +D8 HD with or without RA in terms of late PGC and meiotic genes expression, the +D6 HD was selected as the optimal condition for the following experiments.

### 3.7. The In Vitro Induced Germ Cell-like Cells in +D6 HD EBs Express DDX4 but Not SYCP3 Protein

To confirm that induced hPGC-LCs gained further differentiation in hanging drop EBs, the presence of DDX4 and SYCP3 proteins was investigated by immunofluorescence. In PB12 hiPSCs-derived EBs, DDX4 and SYCP3 were completely absent at day 4 of culture (Figure 7A). However, after an additional 6-day culture in the hanging drop (+D6 HD) we retrieved DDX4-positive cells in EBs without any detection of the SYCP3 protein (Figure 7A). In human fetal ovary, used as a positive control, both proteins were detected (Figure 7B). To assess the robustness of our culture system, the same experiment was repeated using another hiPSC line. Similarly, SMA hiPSC-derived EBs did not show DDX4 on day 4, while this protein was detected in +D6 HD EBs (Appendix A). No SYCP3 expression was detected either in day 4 SMA EBs or in +D6 HD (Appendix A). Quantifications evidenced similar percentages of DDX4-positive cells between PB12 and SMA hiPSC-derived +D6 HD EBs (11.61% vs. 13.09%, *p* = 0.35) (Appendix A). Thus, these results demonstrate that early hPGC-LCs can be differentiated up to the acquisition of gametogenic competency using a hanging drop culture system despite meiotic genes being transcribed but not translated.

### 3.8. Down Expression of POU5F1 and TFAP2C Precedes the Apparition of DDX4-Positive Cells in Hanging Drop EBs

To better understand the kinetics of development of hPGC-LCs in vitro, an immunohistological analysis was performed on EB sections for POU5F1, TFAP2C and DDX4 proteins (Figure 8A). On day 4 EBs, POU5F1- and TFAP2C-positive proteins are present at a mean percentage of 30.12 and 35.11%, respectively, while the DDX4 protein is completely absent (Figure 8B). In +D2 HD, the mean percentages of OCT4-postive and TFAP2C-positive cells are dramatically decreased when compared with day 4 EBs, and DDX4 protein is still undetectable (Figure 8B). Interestingly, while the dramatic decrease of POU5F1 and TFAP2C was maintained between +D2 HD and +D8 HD, the DDX4 protein appeared for the first time in 3.80% of cells per EB section in +D4 HD (Figure 8B). A significant increase in the mean percentage of DDX4-positive cells was observed in +D6 HD when compared with +D4 HD (13.09% vs. 3.80%; *p* = 0.01). Although the DDX4-positive cells maintained a proliferation potential in +D6 HD as they expressed the PCNA marker (Appendix A), DDX4 expression at the protein level disappeared in +D8 HD (Figure 8B). Thus, downregulation of pluripotency precedes the apparition of late PGC markers. Unfortunately, it seems that differentiated hPGC-LC cannot be further maintained in the hanging drop model and likely require a more appropriate niche at this stage.

## 4. Discussion

This work confirms the possibility to efficiently differentiate PGC-LC from human pluripotent stem cells up to the oogonia stage and to further extend our ability to sort and manipulate such cells. This is a step forward towards human oogenesis in vitro, a unique tool to model the pathologies of the female germline such as primary ovarian insufficiency. Here, we optimized hPGC-LC derivation. First, we provided a new tool to sort these cells. Second, we set up a protocol that allows for the fast and efficient production of gametogenic competent cells.

The expression of DAZL and DDX4 in germ cells is believed to sign the acquisition of the gametogenic competency [31,32,33]. Indeed, we observed a clear correlation between the expression of *DAZL* and that of meiotic genes, indicating that cells exhibiting *DAZL* and *DDX4* expression are likely to be on the verge of entering meiosis. This agrees with the *Dazl* KO mice that are unable to express the meiotic program, an observation that positioned DAZL as the key intrinsic factor for acquiring the gametogenic competency [34,35,36]. Interestingly, we proved here that *DAZL* and *DDX4* can appear in cells differentiated outside the gonadal environment.

The efficient derivation of early human PGC-LCs from iPSCs has been reported by several teams with an efficiency up to 30% [14,37]. We here achieved a similar efficiency confirming the robustness of such a protocol. However, to the best of our knowledge, only one report has recently described the further differentiation of hPGC-LCs to the oogonia stage, competent for meiotic entry. In this pioneer work, Yamashiro et al. [16] described the progressive differentiation of hPGC-LCs in a niche formed with fetal mouse somatic ovarian cells after several months in culture. Our data is coherent with this study, as we could not obtain the differentiation of hPGC-LCs in aggregates composed of fetal somatic ovarian cells after 16 (mouse) or 30 days (human). Indeed, Yamashiro et al. [16] observed the apparition of 0.6% of DDX4-positive cells only after 77 days of culture. Therefore, we conclude that the fetal ovarian niche is a poor promoter of hPGC differentiation.

Here we proposed that a simple hanging drop culture system of EBs in the presence of cytokines (BMP4 and SCF) can promote the appearance of up to 13% of DDX4-positive cells in 6 days. This fast differentiation is accompanied by the expression of meiotic genes despite the fact that those cells did not actually initiate meiosis. Interestingly, it was recently reported that BMP4 signaling and its downstream target ZGLP1 are keys to launching the oogenic program prior to meiotic entry both in vivo and in mouse PGC-LCs [38]. We may thus speculate that BMP4 plays a similar role in hPGC-LC and that its presence in hanging drops has prompted germ cell differentiation. Further studies are necessary to investigate BMP4 functions in the human germline. Altogether, the here described model is a promising tool to further progress towards human oogenesis in vitro. However, the main drawback of our system is that it does not allow germ cell maintenance. On the contrary, our data using reconstituted ovaries (aggregates with fetal ovarian somatic cells) proved that in such an environment the germ cells survived well and can be maintained for long periods. This is also in accordance with the work from Yamashiro et al. [16]. Therefore, it seems that the ovarian niche is required for germ cell survival and future work should determine whether combining both models, an efficient differentiation in hanging drop and proper maintenance in the ovarian niche, can allow oogenesis completion. Unfortunately, we could not achieve this step yet, as the sorting of DDX4-positive cells to culture them within an ovarian niche is currently technically out of reach.

A very interesting feature from our work is that we efficiently sorted early hPGC-LCs based on a surface marker. Many of the most achieved derivation protocols of PGC-LC from pluripotent cells are based on the integration in the genome of reporters coding for fluorescent proteins under the control of promoters of germ-specific genes [14,39,40]. While this is a very reliable and convenient tool to set up experimental conditions, it requires a long engineering process that is poorly compatible with screening the potential of pluripotent cells from numerous patients suffering from infertility issues. We demonstrate that using the PDPN antigen that is efficiently recognized by the D2-40 monoclonal antibody allows for the sorting of most, if not all, hPGC-LCs without using a genetic reporter. Such a cell sorting methodology was previously used to sort actual PGC from human fetal gonads [26,41]. The limit of such a methodology is that the PDPN antigen is robustly expressed in early hPGC (i.e., TFAP2C-positive), but progressively disappears when cells progress towards a meiotic fate and thus it cannot be used to reliably sort more differentiated hPGC-LCs.

On a surprising note, although we systematically tested the addition of high doses of RA, we did not observe any significant differentiation induced by RA. This is odd, as RA has been proposed as the signal triggering the expression of STRA8, the gatekeeper of the mitotic/meiotic transition [42]. We and others demonstrated that in the human fetal ovary, RA can provoke some germ cell differentiation and even meiotic entry [30,43]. However, it should be noted that such an effect was quite weak compared to mouse embryonic ovaries. Additionally, it was recently proposed that, in mouse ovaries, RA may not be the key signal initiating meiotic entry [8,44,45]. This would fit with the fact that even in the hanging drop model with DDX4-positive cells, RA did not trigger an increase in *STRA8* expression nor in meiotic genes, apart from a mere amplification of *SYCP3*. We thus believe that the actual meiotic trigger remains to be identified and that this would greatly improve our ability to progress towards the formation of human oocytes in vitro. It should be stressed that in a reconstituted ovary culture of up to 120 days the meiotic entry also appears to be main limiting step [16]. Of course, one cannot exclude the possibility that the aging ovarian niche has lost the ability to trigger meiotic entry or that we could not observe meiosis in the hanging drop due to the inability of the model to sustain germ cell survival. Recent studies reported the in vitro generation of Fetal Ovarian Somatic Cells-like cells (FOSLC) from mouse pluripotent stem cells [46,47] with high similarity to endogenous ovarian somatic cells immediately prior the onset of meiosis (12.5 dpc). When FOSLC were aggregated with mPGC-LC or sorted endogenous PGC, murine germ cells could enter and complete meiosis leading to the formation of functional oocytes [46]. This complete mouse in vitro oogenesis highlights the importance of the matching between germ cells and their environment. Gonen et al. [48] used a similar approach to generate gonadal somatic cell progenitors using hiPSC. This appears to be a promising tool compared to the use of endogenous ovarian somatic cells, whose use relies on poorly accessible biological material. Its optimization may allow the production of a developmentally fitted gonadal environment to support both hPGC-LC survival and maturation in a controlled way.

## 5. Conclusions

Overall, we believe our work provides new tools to progress towards realizing human oogenesis in vitro. Future works will have to address the reliability of such artificial gametogenesis through an extensive evaluation, both at transcriptomic and epigenomic levels. The current main limitation to further progress towards such a goal appears to be our lack of knowledge of the human embryonic female germline and of the key regulators of meiosis [49,50], a central event during gametogenesis.

## Figures and Tables

**Figure 1 cells-11-03832-f001:**
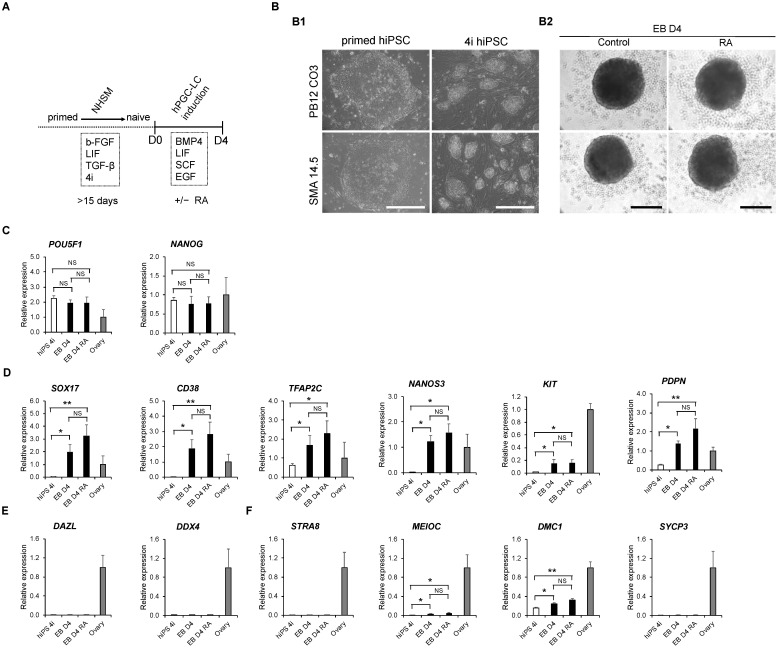
In vitro induction of early, but not late, hPGC-LC from hiPS cells. (**A**) Schematic illustration of in vitro specification procedure of hPGC-LC from hiPSCs. (**B**) In vitro day 4 embryoid body formation from naïve hiPSCs in the absence or presence of RA. (**B1**): hiPS colony images in primed (large flat-shaped colony) and naïve (small domed-shaped colonies) states of pluripotency. (**B2**): embryoid body images with or without RA. Scale bars, 500 µm or 200 µm, respectively. (**C–F**) Gene expression analysis of day 4 embryoid bodies by RT-qPCR with or without RA. The expression of pluripotency (**C**), early hPGC (**D**), late hPGC (**E**) and meiotic (**F**) gene markers are illustrated. The results are presented as the mean ± sem., with *n* = 4 (4i hiPS); *n* = 5 (EB D4); *n* = 6 (EB D4 RA) and *n* = 3 (7.8 to 10 GW human fetal ovary). Asterisk indicates a statistically significant difference (*: *p* < 0.05; **: *p* < 0.01). 4i: Four Inhibitory Molecules for naïve-like state of pluripotency; EB D4: Embryoid Body at Day 4; EB D4 RA: Embryoid Body at Day 4 with Retinoic Acid; NHSM: Naïve Human Stem cell Medium; NS: Not statistically Significant; RA: Retinoic Acid.

**Figure 2 cells-11-03832-f002:**
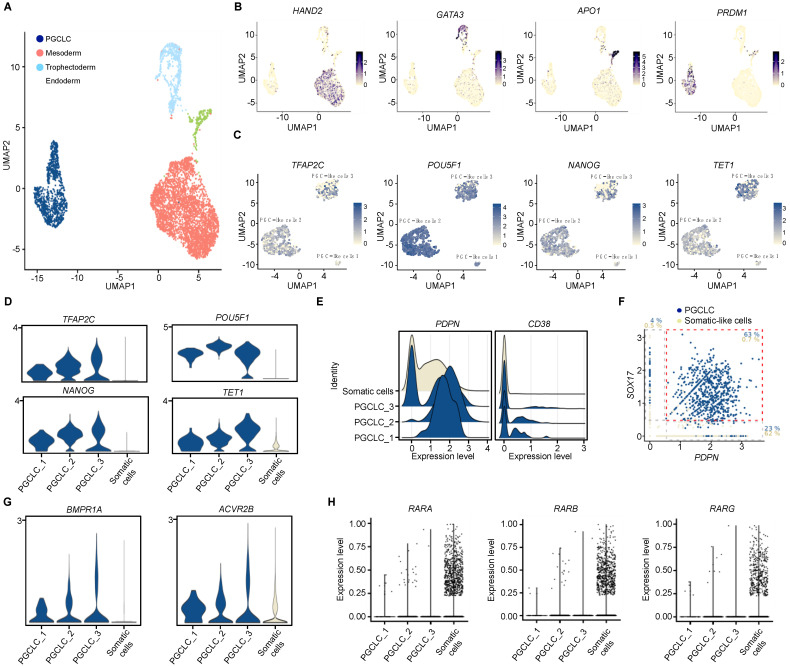
Transcriptional characterization of day 4 EB PGC-LC using scRNA-Seq. (**A**) Two-dimensional UMAP (Uniform Manifold Approximation and Projection) plot of the 5421 cells from two pools of day 4 EB. Cell clusters are colored according to the cell identity (mesoderm, endoderm, trophectoderm or PGC-LC lineages). (**B**) Projection of the expression level of *HAND2* (mesoderm), *GATA3* (trophectoderm), *APO1* (endoderm) and *PRDM1* (PGC). (**C**) Projection of the expression level of the PGC (*TFAP2C*, *NANOG*, and *POU5F1*) and the epigenetic (*TET1*) marker genes in the identified three sub-clusters of PGC-LCs. (**D**) Violin plots showing the expression of *TFAP2C*, *NANOG*, *POU5F1* and *TET1* genes in day-4 PGC-LCs compared to somatic cells. (**E**) Ridge Plot showing the expression levels of *PDPN* and *CD38* genes in comparison to the cell density in different PGC-LC and somatic cell clusters. (**F**) Scatter plot shows the co-expression correlation of *PDPN* with the *SOX17* PGC marker. The percentages of the cell types co-expressing or not *SOX17* and *PDPN* genes are shown either in blue (PGC) or grey (somatic) (**G**) Violin plots for BMP signaling signature in PGC-LCs compared to somatic cells. (**H**) Violin plots showing the gene expression levels of retinoic acid receptors in the indicated conditions.

**Figure 3 cells-11-03832-f003:**
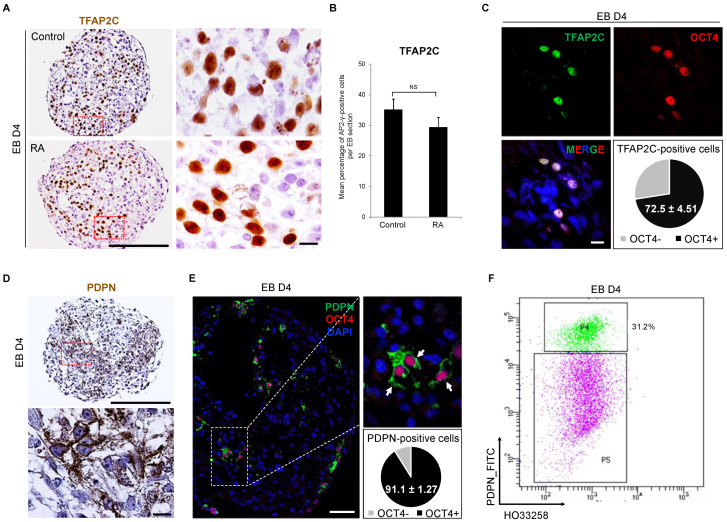
Detection of early hPGC protein markers in day 4 embryoid bodies. (**A**) Immunodetection of TFAP2C protein in a day 4 embryoid body in the absence (top row) or presence (bottom row) of RA. Scale bar, 200 µm. The red dotted box of each condition is enlarged on the right image and corresponds to a ×1000 magnification. Scale bar, 10 µm. (**B**) Percentages of TFAP2C-positive cells per embryoid body section at day 4. The results are presented as the mean ± sem., with *n* = 6 (Control or RA). (**C**) Confocal analyses of TFAP2C and OCT4 co-expression in PGC-LC. The pie chart shows the percentage of OCT4-positive cells within the TFAP2C-positive cells. The results are shown as the mean ± sem., with *n* = 4. (**D**) Immunodetection of the early hPGC surface marker PDPN in a day 4 embryoid body. Scale bar, 200 µm. The red dotted box is enlarged on the right image and corresponds to a ×1000 magnification. Scale bar, 10 µm. (**E**) Confocal analyses of PDPN and OCT4 co-expression in PGC-LC. The white dotted box is enlarged on the right image. Scale bar, 20 µm. The pie chart shows the percentage of OCT4-positive cells within the PDPN-positive cells. The results are shown as the mean ± sem., with *n* = 4. (**F**) Dot plot analyses showing PDPN-positive PGC-LCs population from day 4 dissociated-embryoid bodies. P4 and P5 are the populations of cells considered as PDPN-positive and -negative, respectively. EB D4: Embryoid Body at Day 4; NS: Not statistically Significant; RA: Retinoic Acid.

**Figure 4 cells-11-03832-f004:**
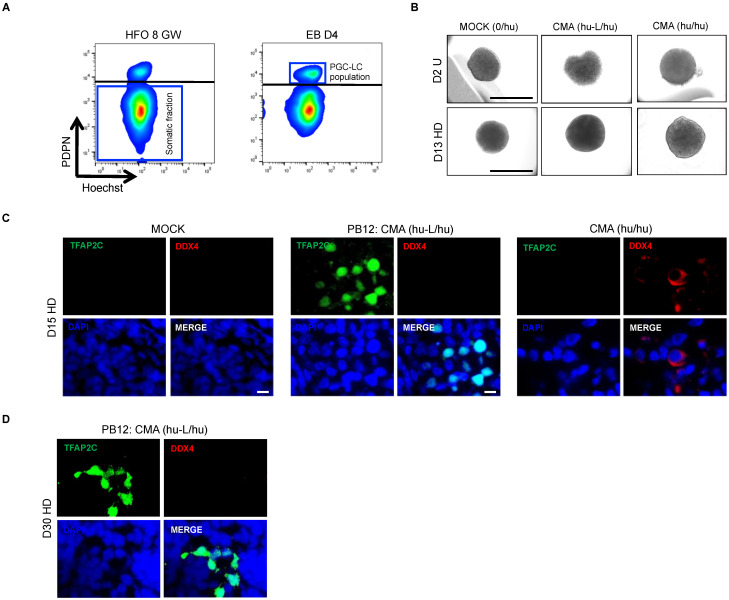
In vitro assessment of early hPGC-LC development when cultured in a somatic environment of human fetal ovary. (**A**) Cytometry plots show the somatic and germ cell populations from a human fetal ovary (plot on the left) and the PGC-LC fraction obtained from day 4 EBs (plot on the right). The day 4 EBs PDPN-positive cells were mixed with the ovarian PDPN-negative cells at a ratio of 2:8 (**B**) Images of day 2 (top row) and day 15 (bottom row) of re-aggregated ovaries in the absence (MOCK) or presence of the PDPN-positive human PGC-LCs [CMA (hu-L/hu)]. A positive control was used by mixing authentic PDPN-positive hPGCs with the ovarian somatic cells population [CMA (hu/hu)]. Scale bar, 500 µm. (**C**) Immunofluorescence analyses for the absence or presence of TFAP2C and DDX4 proteins in MOCK, CMA (hu-L/hu) and CMA (hu/hu) conditions at day 15 of culture. (**D**) Immunofluorescence analyses for the absence or presence of TFAP2C and DDX4 proteins CMA (hu-L/hu) condition at day 30 of culture. CMA hu/hu: Cell Mixture Aggregate (human PGCs/human somatic cells); CMA hu-L/hu: Cell Mixture Aggregate (human PGC-*like* Cells/human somatic cells); D15 HD: Day 13 from Hanging Drop; D30 HD: Day 28 from Hanging Drop; D2 U: Day 2 from U-bottom low adherence 96 well plate; EB D4: Embryoid Body at Day 4; HFO 8 GW: Human Fetal Ovary of 8 Gestational Week; MOCK (0/hu): 8 gestational week human ovarian somatic cells.

**Figure 5 cells-11-03832-f005:**
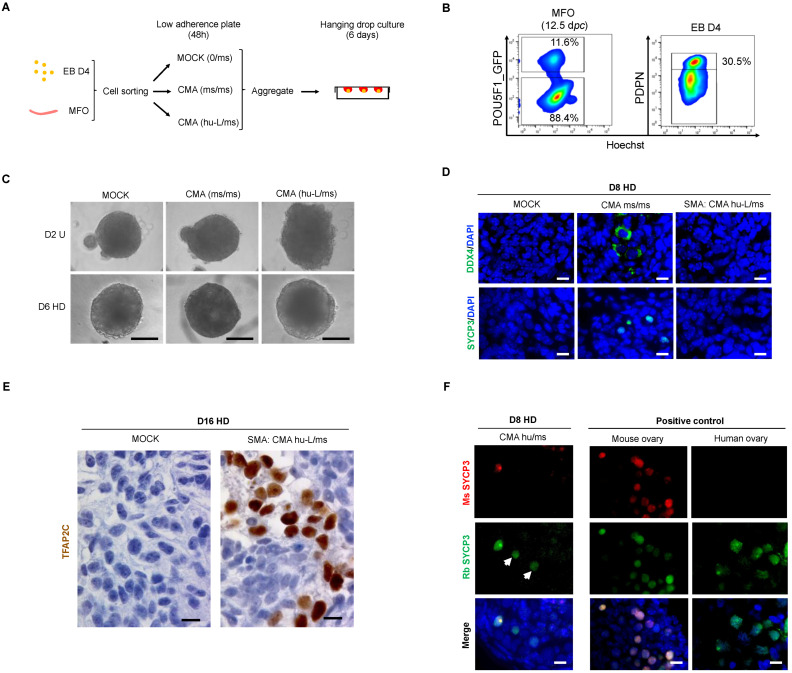
In vitro assessment of early hPGC-LC development when cultured in a somatic environment of murine ovary. (**A**) Schematic representation of in vitro culture procedure of re-aggregated somatic cells of murine ovary with mPGCs [CMA (ms/ms)] or hPGC-LC [CMA (hu-L/ms)] or without mPGCs/hPGC-LC [MOCK (0/ms)]. The Day 2 aggregates were transferred on a hanging drop culture system for 6 days. (**B**) Flow cytometry plots obtained after enzymatic dissociation of 12.5 *dpc* mouse fetal ovaries (left) and day 4 human embryoid bodies (right). Murine PGCs (POU5F1-positive) and ovarian somatic cells (OCT4-negative) were co-sorted with early PDPN-positive hPGC-LC from day 4 embryoid bodies. (**C**) Images of day 2 (top row) and day 8 (bottom row) re-aggregated ovaries in the absence (MOCK) or presence of (i) POU5F1-postive mPGCs [CMA (ms/ms)], and (ii): PDPN-positive hPGC-LC [CMA (hu-L/ms)]. Scale bar, 200 µm. (**D**) Immunofluorescence analyses for the absence or presence of DDX4 and SYCP3 proteins in MOCK, CMA (ms/ms) and CMA (hu-L/ms) conditions at day 8 of culture. (**E**) Immunofluorescence analysis for the detection of SYCP3 protein in CMA (hu/ms). Mouse and human fetal ovaries were used as positive controls for mouse and rabbit SMC3 antibodies. (**F**) Immunohistochemistry for TFAP2C protein detection in CMA hu-L/ms at day 16 of culture. Scale bar 10 µm (**D**–**F**). CMA hu/ms: Cell Mixture Aggregate (human PGC/Mouse somatic cells); CMA hu-L/ms: Cell Mixture Aggregate (human PGC-like Cells/Mouse somatic cells); CMA ms/ms: Cell Mixture Aggregate (Mouse PGC/Mouse somatic cells); D8 HD: Day 6 from Hanging Drop; D16 HD: Day 14 from Hanging Drop; D2 U: Day 2 from U-bottom low adherence 96 well plate; d*pc*: day *post-coïtum*; EB D4: Embryoid Body at Day 4; MFO: Mouse Fetal Ovary; MOCK (0/ms): 12.5 d*pc* mouse ovarian somatic cells only.

**Figure 6 cells-11-03832-f006:**
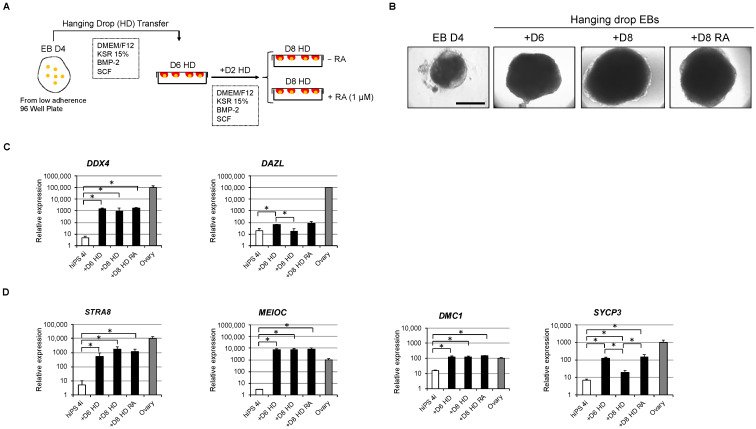
In vitro overexpression of late hPGC-LC and meiotic gene markers in hanging drop-transferred embryoid bodies. (**A**) Schematic illustration of in vitro culture procedure of day 4 embryoid bodies in the hanging drop culture system. At day 10, embryoid bodies are exposed or not to RA for 48 h. (**B**) Images of (i) a day 4 EB in a U-bottom low adherence 96-well plate, (ii) a day 10 EB, and (iii) day 12 EBs in the absence or presence of RA. Scale bar, 500 µm. (**C**,**D**) Gene expression analyses by RT-qPCR of late PGC (**C**) and meiotic gene (**D**) markers in day 0 hiPSC 4i (*n* = 4), day 10 EBs (+D6 HD; *n* = 3), day 12 EBs in the absence or presence of RA (+D8 HD or +D8 HD RA; *n* = 3 for both) and human fetal ovary (*n* = 3), * *p* < 0.05. +D6 HD: +Day 6 in hanging drop; +D8 HD: +Day 8 in hanging drop; +D8 HD RA: +Day 8 in Hanging drop with Retinoic Acid; EB D4: Embryoid Body at Day 4.

**Figure 7 cells-11-03832-f007:**
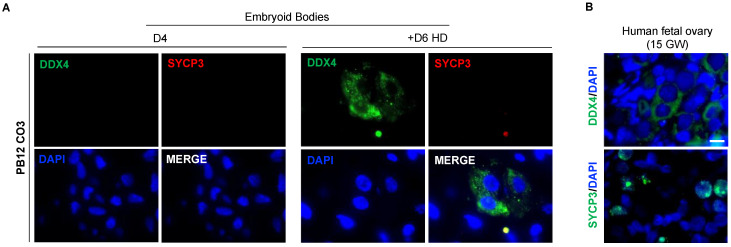
Assessment of late PGC and meiotic proteins expression in PB12 hiPSCs-derived EBs at day 4 and day 10. (**A**) Immunofluorescence analyses of DDX4 and SYCP3 proteins expression in PB12 hiPSCs-derived EBs at day 4 (left panel) and day 10 (right panel). Scale bar, 10 µm. (**B**) Immunofluorescence detection of DDX4 and SYCP3 proteins in 15 GW human fetal ovary, used as a positive control. Scale bar, 10 µm. +D6 HD: +Day 6 in hanging drop; EB D4: Embryoid Body at Day 4; GW: Gestational Week.

**Figure 8 cells-11-03832-f008:**
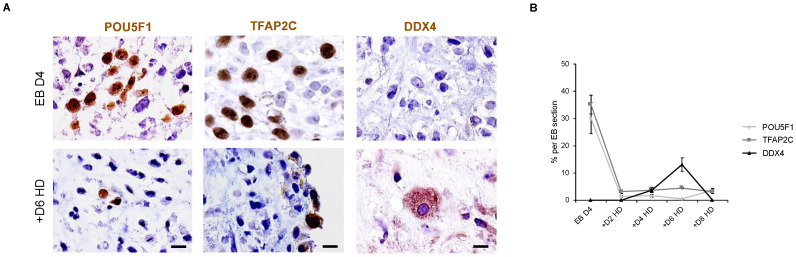
Assessment of kinetic expression of pluripotency, early PGC and late PGC protein markers in embryoid bodies before and after hanging drop culture. (**A**) Immunodetection of POU5F1, TFAP2C, PDPN and DDX4-positive cells in day 4 or day 10 embryoid bodies. All images are at a ×1000 magnification with a scale bar of 10 µm. (**B**) Percentages of POU5F1, TFAP2C, PDPN and DDX4-positive cells per embryoid body section at days 4, 6, 8, 10 and 12. The results are presented as the mean ± sem., with *n* = 3. Conditions that do not share a common letter are significantly different (*p* < 0.05). +D2 HD: +Day 2 in Hanging drop; EB D4: Embryoid Body at Day 4.

## Data Availability

The single-cell RNA-seq data presented in this study are openly available in the European Bioinformatics Institute (EMBL-EBI) under the accession numbers e-mtab-12089.

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
