# Peer review of "Sorting and Manipulation of Human PGC-LC Using PDPN and Hanging Drop Cultures"

_cells, 2022, doi:10.3390/cells11233832_

Round 1

Reviewer 1 Report

Brahim Arkoun and colleagues reported an in vitro system that induced human iPSCs to differentiate into PGC-like cells, which could further differentiate into oogoina stage and expressed germline marker DDX4 and meiotic markers. This study is of importance, and it provides new reference for clinic research. The quality of this manuscript is good. This manuscript is acceptable, after revision of two minor issues.

1.     To characterize early hPGC-LC/oogoina, the proliferation potential should be detected. PCNA staining or dual IF staining of BrdU/DDX4 are recommended to determine the proliferation potential of hPGC-LC.

2.     The coculture assays revealed a detrimental effect of human ovarian somatic cells on hPGC-LC differentiation, since marker proteins such as DDX4 and SYCP3 were not detected. However, there is a possibility that hPGC-LC needs longer time for differentiation in the microenvironment composed of human ovarian somatic cells, compared to in mouse ovarian somatic cells. In this case, it’s recommended to determine the expression of DDX4 or Sycp3 at mRNA level in these hPGC-LC, to find out whether the transcription initiates. If yes, it indicates that some key factors are probably required to further promote differentiation, and this may enhance the significance of this study in clinic research.

Author Response

Point-by-point response to reviewers 

We thank the Reviewers for their constructive comments and questions. We have addressed them experimentally by performing additional analyses and below with a point-by-point response. We also prepared a revised version of the manuscript in which the modified sentences are highlighted in yellow. English writing has been improved by a scientific native speaker. We believe that the review process has significantly strengthened our conclusion, as well as the clarity of the manuscript in general. We hope that you will appreciate these responses and now find it suitable for publication. 

Reviewer 1: 

Brahim Arkoun and colleagues reported an in vitro system that induced human iPSCs to differentiate into PGC-like cells, which could further differentiate into oogoina stage and expressed germline marker DDX4 and meiotic markers. This study is of importance, and it provides new reference for clinic research. The quality of this manuscript is good. This manuscript is acceptable, after revision of two minor issues. 

We greatly appreciate the Reviewer's kind general comments and support. 

1. To characterize early hPGC-LC/oogonia, the proliferation potential should be detected. PCNA staining or dual IF staining of BrdU/DDX4 are recommended to determine the proliferation potential of hPGC-LC. 

We agree with the Reviewer and appreciate the opportunity to refine our result. 

Therefore, we first assessed the proliferation potential of early hPGC-LC from day 4 EB using Ki67 marker. The results are presented in the new Figure S2 and integrated into the “Results” section as follows: 

“To verify the proliferation potential of these early hPGC-LCs, the expression of Ki67 marker was assessed. More than half of the TFAP2C-positive cells were also Ki67-positive (Figure S2) indicating that hPGCLC divide actively.” 

We verified the expression of PCNA marker within the DDX4-positive cells from +D6 HD. The result is now available in new Figure S7. However, due to limited material and antibodies incompatibility we did not succeed yet to analyze the proliferation potential of DDX4-positive cells with sufficient accuracy to propose a neat quantification. and integrated into the result section as follows: “Altough the DDX4-positive cells maintained a proliferation potential in +D6 HD as they expressed the PCNA marker (Figure S7), DDX4 expression at the protein level disappeared in +D8 HD (Figure 8B).” 

2. The coculture assays revealed a detrimental effect of human ovarian somatic cells on hPGC-LC differentiation, since marker proteins such as DDX4 and SYCP3 were not detected. However, there is a possibility that hPGC-LC needs longer time for differentiation in the microenvironment composed of human ovarian somatic cells, compared to in mouse ovarian somatic cells. In this case, it’s recommended to determine the expression of DDX4 or Sycp3 at mRNA level in these hPGC-LC, to find out whether the transcription initiates. If yes, it indicates that some key factors are probably required to further promote differentiation, and this may enhance the significance of this study in clinic research. 

We agree with the relevant comment of the Reviewer and we acknowledge the importance to verify whether DDX4 is expressed at the mRNA level in hPGCLC cocultured with fetal ovarian somatic cells. 

Unfortunately, all co-culture aggregates (hPGC-LC/human fetal ovarian somatic cells) performed in this study were fixed and intended for immunohistological analyses. No RNA from these rare aggregates is thus available. Additionally, because of the rarity of fetal ovaries and subsequently of aggregates, we think it would be very hard to reach sufficient amounts of total RNA to analyze meiotic genes expression by qPCR. Furthermore, the expression of DAZL and DDX4 marks the gametogenic licensing (i.e. the potential for expressing meiotic genes). With no DDX4 protein in aggregates, if meiotic genes were expressed it would be at extremely low level. For these reasons, we consider that such an experiment is poorly feasible and might not be easy to interpret. 

Reviewer 2 Report

In this manuscript, the author reported a novel marker to sort the human primordial germ like cells without the undesirable genetic modifications. The authors further demonstrated a fast and efficient method to differentiate hPGC-LC to gametogenic competent cells. The authors have conducted comprehensive studies to identify the novel marker and characterizing the early and late markers. I have only minor corrections to suggest.

Minor changes

1.       Figure correction: Figure 2A) Endoderm marker in legend missing

2.       Line 331-333: How is this contrasting with scRNA seq-data? The scRNA seq data also suggest a very few numbers of somatic cells positive for PDPN. What is the percentage of those cells from scRNA seq?

3.       Section 3.7: The SYCP3 transcripts were observed but no translation. Have you considered increasing the number of days in culture to see translation? Have you checked other genes of the synaptonemal complex (SYCP1, SYCP2, SYCE1, SYCE2, SYCE3, TEX12, SIX6OS1?

4.       Line 554-555: References missing

5.       Recent paper not cited: Abdyyev VK, Sant DW, Kiseleva EV, Spangenberg VE, Kolomiets OL, Andrade NS, Dashinimaev EB, Vorotelyak EA, Vasiliev AV. In vitro derived female hPGCLCs are unable to complete meiosis in embryoid bodies. Exp Cell Res. 2020 Dec 15;397(2):112358. doi: 10.1016/j.yexcr.2020.112358. Epub 2020 Nov 5. PMID: 33160998. This paper concludes that hPGC-LCs alone cannot differentiate into haploid cells without the supporting cells of the gonad.

Author Response

 Point-by-point response to reviewers 

We thank the Reviewers for their constructive comments and questions. We have addressed them experimentally by performing additional analyses and below with a point-by-point response. We also prepared a revised version of the manuscript in which the modified sentences are highlighted in yellow. English writing has been improved by a scientific native speaker. We believe that the review process has significantly strengthened our conclusion, as well as the clarity of the manuscript in general. We hope that you will appreciate these responses and now find it suitable for publication. 

Reviewer 2: 

In this manuscript, the authors reported a novel marker to sort the human primordial germ like cells without the undesirable genetic modifications. The authors further demonstrated a fast and efficient method to differentiate hPGC-LC to gametogenic competent cells. The authors have conducted comprehensive studies to identify the novel marker and characterizing the early and late markers. I have only minor corrections to suggest. 

We deeply thank the Reviewer for the very kind general comment. 

Minor changes 

1. Figure correction: (Figure 2A) Endoderm marker in legend missing 

We agree with the Reviewer. The endoderm marker in legend has been added in the Figure 2A. 

2. Line 331-333: How is this contrasting with scRNA seq-data? The scRNA seq data also suggest a very few numbers of somatic cells positive for PDPN. What is the percentage of those cells from scRNA seq? 

Quantification of somatic-like cells expressing PDPN mRNA (Figure 2E) indicates that 62.8 % of these cells expressed PDPN in scRNA-seq data. Immunostaining with the M2A antibody revealed about 1% of somatic cells expressing PDPN. On the other hand mRNA and protein are observed in 86.5% and 98% of hPGC-LC respectively. Such a contrast may be due to many reasons: specific isoforms, no translation, no membrane targeting … As these hypotheses are speculative, we did not further comment this point in the manuscript and merely clarify the contrast : “On the other hand, very few somatic-like cells presented a staining for PDPN (1.01±0.20 % of POUF1-negative cells were PDPN-positive, n=4) indicating a specific presence of the protein at the membrane of hPGC-LC and contrasting with scRNA-seq data (62.8% of somatic-like cells expressing the PDPN mRNA, Figure 2E)” 

3. Section 3.7: The SYCP3 transcripts were observed but no translation. Have you considered increasing the number of days in culture to see translation? 

We agree with the Reviewer. Indeed, we performed longer time of culture until day 8 in hanging drop as shown in figure 8B; Unfortunately, while the percentage of VASA-positive cells was increasing between day 2 and day 6, these cells were completely lost at day 8 indicating that these may need a specific environment to sustain their survival. (see line 519-521). 

Have you checked other genes of the synaptonemal complex (SYCP1, SYCP2, SYCE1, SYCE2, SYCE3, TEX12, SIX6OS1? 

We thank the Reviewer for this suggestion and appreciate the opportunity to investigate other synaptonemal complex genes. 

Indeed, after we verified the expression of SYCP2, SYCE1 and SYCE3 genes by qPCR, they were significantly increased in hanging drop culture at day 6 when compared with day 4 EB or 4i hiPSCs. These results are now illustrated in new Figure S5E and described in the result section as follows: 

“Additionally, the expression of the synaptonemal complex genes significantly increased at day 6 in hanging drop when compared with 4i hiPSCs or day 4 EBs (Figure S5E).” 

4. Line 554-555: References missing 

We agree with the Reviewer and two references were added: Irie et al., 2015 and Sazaki et al., 2015, therefore it becomes: “The efficient derivation of early human PGC-LCs from iPSCs has been reported by several teams with an efficiency up to 30% (Irie et al., 2015, Sasaki et al., 2015)”. 

5. Recent paper not cited: Abdyyev VK, Sant DW, Kiseleva EV, Spangenberg VE, Kolomiets OL, Andrade NS, Dashinimaev EB, Vorotelyak EA, Vasiliev AV. In vitro derived female hPGCLCs are unable to complete meiosis in embryoid bodies. Exp Cell Res. 2020 Dec 15;397(2):112358. doi: 10.1016/j.yexcr.2020.112358. Epub 2020 Nov 5. PMID: 33160998. This paper concludes that hPGC-LCs alone cannot differentiate into haploid cells without the supporting cells of the gonad. 

We thank the Reviewer for this relevant suggestion and the reference is now cited in the Introduction section as: “Recently, it was reported that hPGC-LCs specified from iPSCs failed to complete meiosis in the absence of the supporting gonadal niche (Abdyyev et al., 2020)”. 

Round 2

Reviewer 1 Report

After revision, this manuscript is acceptable.